# Cross-View Consistency Regularisation for Knowledge Distillation

## ABSTRACT

Knowledge distillation (KD) is an established paradigm for transferring privileged knowledge from a cumbersome model to a more lightweight and efficient one. In recent years, logit-based KD methods are quickly catching up in performance with their feature-based counterparts. However, existing research has pointed out that logit-based methods are still fundamentally limited by two major issues in their training process, namely overconfident teacher and confirmation bias. Inspired by the success of cross-view learning in fields such as semi-supervised learning, in this work we introduce within-view and cross-view regularisations to standard logit-based distillation frameworks to combat the above cruxes. We also perform confidence-based soft label selection to improve the quality of distilling signals from the teacher, which further mitigates the confirmation bias problem. Despite its apparent simplicity, the proposed Consistency-Regularisation-based Logit Distillation (CRLD) significantly boosts student learning, setting new state-of-the-art results on the standard CIFAR-100, Tiny-ImageNet, and ImageNet datasets across a diversity of teacher and student architectures, whilst introducing no extra network parameters. Orthogonal to on-going logit-based distillation research, our method enjoys excellent generalisation properties and, without bells and whistles, boosts the performance of various existing approaches by considerable margins. Our code and models will be released.

## CCS CONCEPTS

• **Computing Methodologies → Machine Learning**.

## KEYWORDS

Knowledge Distillation, Neural Networks, Image Classification

**ACM Reference Format:**
Anonymous Author(s). 2024. Cross-View Consistency Regularisation for Knowledge Distillation. In *Proceedings of 32nd ACM International Conference on Multimedia (MM '24)*. ACM, New York, NY, USA, 10 pages. https://doi.org/XXXXXXX.XXXXXXX

## 1 INTRODUCTION

Deep neural networks (DNNs) have achieved tremendous success across a plethora of computer vision, natural language processing, and multimedia tasks [17, 21, 44]. Behind their widespread applications, high-performance DNNs are often associated with larger if

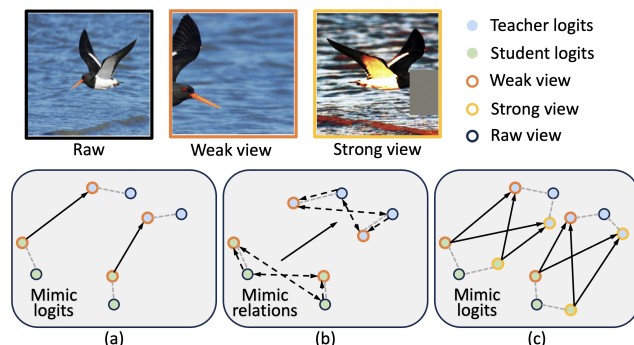

**Figure 1: A schematic comparison of logit-based distillation methods from a cross-view learning perspective. (a): Methods mimicking logits in an unitary view [10, 24, 28, 31, 36, 60, 69]. (b) Methods optimising and mimicking contrastive relations [54]. (c) Our CRLD which involves within-view and cross-view logit transfer.**

not prohibitive model sizes and computational overheads, which render them hard to implement on resource-constrained devices and platforms. Towards computation-efficient, storage-friendly, and real-time deployment of DNNs, a viable solution is knowledge distillation (KD), which was first proposed by Hinton et al. [24] for model compression. KD works by transferring the advanced capability of a larger, cumbersome teacher model to a more lightweight and efficient student model. Since its proposal, KD has witnessed significant advancements in the past decade as a range of feature-based [7, 8, 20, 23, 34, 38, 41, 49, 50, 59, 61, 64] and response-based (logit-based) [10, 24, 28, 36, 54, 60, 69] KD algorithms are proposed for diverse tasks and applications [24, 25, 35, 52, 58, 71]. State-of-the-art KD methods have largely reduced the teacher-student performance gap. For instance, top-performing methods [10, 28, 69] are capable of training students that are on par with or even surpass their corresponding teacher models on smaller datasets such as CIFAR-100 (see Table 1), and are not far behind on larger datasets [16] (Tables 3 and 4).

In this paper, our goal is to further advance the capability of knowledge distillation by addressing two long-standing problems in existing KD methods. Previous research has reported that stronger teacher models and more accurate teacher predictions do not necessarily lead to better distilled students [11, 24, 36, 62]. This counter-intuitive observation points to a prominent and fundamental problem in knowledge distillation — **overconfident teacher**. In the pioneering work of KD [24], Hinton et al. argued that valuable information is hidden in teacher's predictions of the non-target classes. These predictions, known as the "dark information", are however largely suppressed when the teacher make predictions with an overly-high confidence. Hence, regularisation of teacher

predictions is essential to distilling knowledge with greater generalisation capabilities to the student [18, 24, 37].

In their work [24], Hinton et al. propose to mitigate the overconfidence problem by softening the predicted probabilities after Softmax using the temperature hyperparameter. This practice is inherited by many later works [10, 28, 31, 47, 60, 69]. Some methods [36] produce smoothed teacher predictions by introducing auxiliary teacher networks with smaller capacity. More straightforward techniques have also been investigated, including label smoothing [37] and early stopping [37]. These works also highlighted overfitting as another detrimental phenomenon closely related to overconfident teacher. These efforts motivate us to look at consistency regularisation via view transformation — another viable solution to combat overconfidence and overfitting. Although widely explored in the semi-supervised learning (SemiSL) literature [45, 53], consistency regularisation and view transformation have received little attention in knowledge distillation research. According to [57], strong augmentation amplifies the dark information that is insignificant in the weak view. As such, in this paper we reframe these techniques for KD by designing a novel set of within-view and cross-view consistency regularisation objectives and achieve state-of-the-art KD performance.

On the other hand, teacher's predictions are not always correct. **Confirmation bias** [2] arises when erroneous pseudo-labels predicted by the teacher is used to teach the student. In existing logit-based methods [10, 24, 31, 60, 69], the student is designated to faithfully learn whatever supervision the teacher has to provide. Such blind mimicking neglects a key fact that the teacher's predictions may be erroneous and misleading, thereby exacerbating the confirmation bias phenomenon. It has been pointed out in recent research [57] that strong perturbation helps mitigate such confirmation bias, which also supports our introduction of a strongly augmented view of the input image. To further mitigate confirmation bias, we draw inspiration from state-of-the-art SemiSL frameworks whose success is partially attributed to their confidence-aware pseudo-labelling [45, 53, 57]. As such, we propose to selectively pick the more reliable predictions made by the teacher for the student to learn, which is proven beneficial in our experiments.

The considerations and designs described above altogether lead to a novel *Consistency-Regularisation-based Logit Distillation* framework, dubbed "CRLD". By drawing inspiration and reaping the fruits from orthogonal research on semi-supervised learning (SemiSL), CRLD presents a simple yet highly effective and versatile solution to knowledge distillation. It easily boosts advanced KD methods, including DKD [69], MLLD [28], and NormKD [10], without introducing extra network parameters. When applied to existing top-performing methods [10, 28], CRLD establishes new state-of-the-art results across different datasets. A schematic comparison of CRLD against prior logit-based approaches from a cross-view learning perspective is presented in Figure 1

In summary, the contributions of this paper include:

(1) We introduce extensive within-view and cross-view consistency regularisation to combat the overconfident teacher and over-fitting problems common in KD.

(2) We design a reliable pseudo-label mining module to avoid the negative impact of unreliable and erroneous supervisory signals from the teacher, thereby mitigating the confirmation bias in KD.

(3) We present the simple, versatile, and highly effective CRLD framework. CRLD achieves new state-of-the-art results on multiple benchmarks across diverse network architectures and readily boosts existing logit-based methods by considerable margins.

## 2 RELATED WORK

### 2.1 Knowledge Distillation

Knowledge distribution (KD) is first proposed in [24] as a model compression technique. It transfers advanced knowledge from a larger, cumbersome "teacher" model to a smaller, lightweight "student" model. Following its initial success in image classification [24] and object detection [52, 58, 70], KD has quickly swept over many more challenging tasks in 3D scene understanding [19, 25, 66, 71]. Existing KD methods are primarily divided into feature-based and logit-based distillation according to where in the network knowledge transfer takes place — the feature space or the logit space.

**Feature-based Distillation**. As its name suggests, feature-based distillation transfers knowledge in the intermediate feature space of the teacher and student models. A most straightforward way is to simply let the student mimic the features of the teacher, as is done in many early works [1, 23, 41, 49]. Some methods also mine and transfer higher-order information from the teacher's feature maps to the student, including inter-channel [34], inter-pixel [50], inter-layer [61], inter-class[27], intra-class[27], and inter-sample [38–40] correlations, inter-sample distance [38], as well as the teacher network's attention [20, 64]. Generative modelling has also been leveraged for feature-based distillation [59], where randomly masked student features are required to re-generate full teacher features. In addition, cross-stage distillation paths [7, 8] and one-to-all pixel paths [32] have also been proposed for improved feature-based distillation.

**Logit-based Distillation.** Logits are the prediction output by a neural network before its final Softmax layer. Distillation methods that perform knowledge transfer in the prediction space are referred to as logit-based or response-based distillation. Pioneering methods KD [24] and DML [68] directly transfer the teacher's predictions to the student by minimising the Kullback-Leibler (KL) divergence between their predictions. Similar to advances in feature-based distillation, intra-sample and inter-sample relations are also exploited for transfer within the logit space in several works [28, 54]. Instead of treating all logits indifferently, DKD [69] and NKD [60] decompose all logits into target-class and non-target class logits and treat them separately and demonstrate higher knowledge transfer performance. More recent methods such as CTKD [31] and NormKD [10] adopt dynamic Softmax temperature, as opposed to a de facto fixed temperature value in previous works [24, 36, 68, 69], reporting state-of-the-art performance. Another branch of methods [36, 46] introduce assistant networks between the teacher and the student to aid the former's imparting of logit-space knowledge to the latter.

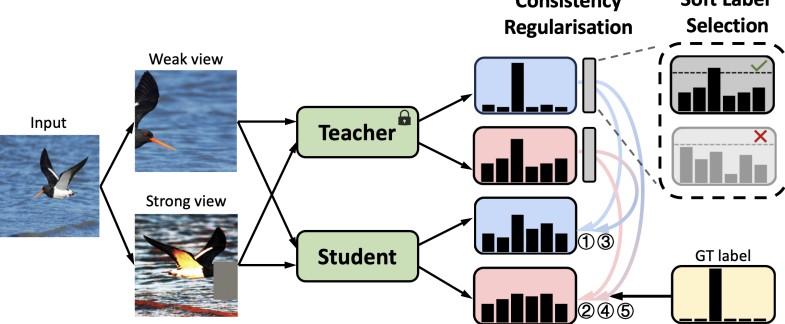

**Figure 2: The CRLD framework. An input image is transformed into a weak view and a strong view, both fed into the teacher and the student separately, yielding four predictions of the same instance. Amongst them, two types of consistency regularisation are enforced: within-view (①②) and cross-view (③④) consistency. Besides, student's prediction of the weak view is supervised by ground-truths (⑤) as per standard practice.**

## 2.2 Consistency Regularisation

Consistency regularisation is at the core of the recent success of state-of-the-art semi-supervised learning algorithms [3, 4, 30, 42, 45, 48, 53]. It involves enforcing invariant representations across different views of the same unlabelled input image to improve the generalisation of learnt representations on unseen data and distribution. The different views of an input image are generated by semantic-preserving transformations, from simple operations such as random crop, horizontal flip, and MixUp [65] as weak transformations, to more sophisticated [13, 22] or even adaptive [3, 12] augmentation strategies for producing strongly augmented views. Given these artificially generated views, representation consistency can be enforced across two stochastical weak views as is done in [4, 53], or a pair of strong and weak views as in [3, 45]. To our best knowledge, beyond SemiSL, the idea of consistency regularisation has not been explored within the context of knowledge distillation.

## 2.3 Data Augmentation for KD

Data augmentation has served as a main pillar to the triumph of deep learning. It transforms training samples into augmented versions whilst preserving their semantic connotation. This results in an abundant if not unlimited amount of extra training data to improve the generalisation of deep neural networks. In the context of knowledge distillation, data augmentation is yet to receive considerable attention, and only a handful of preliminary studies are conducted [5, 14, 15, 51]. Specifically, Das et al. [15] studies the effect of data augmentation in training the teacher model. SSKD [54] and HSAKD [55] incorporate elements of contrastive learning. They apply simple rotation to establish a self-supervised pretext task for improved student learning. Wang et al. [51] and IDA [14] design data augmentation strategies tailored to to boost KD. In contrast, our focus is not the design of a data augmentation strategy itself but to leverage the idea of consistency regularisation to improve student's learning. Data augmentation is simply our tool to allow the formulation of consistency regularisation objectives, as is done in advanced SemiSL methods, and we simply use RandAugment as our strong view transformation following them.

## 3 METHODOLOGY

### 3.1 Knowledge Distillation

Knowledge distillation (KD) involves the student model learning from both ground-truth labels and distillation signals from a pretrained teacher. For the image classification task, ground-truth supervisions are widely enforced via a cross-entropy minimisation objective $\mathcal{L}_{CE}$; the distillation objective $\mathcal{L}_{KD}$ is enforced by minimising the distance between either the intermediate features or the final predictions of the teacher and the student. Thus, KD in its simplest form has $\mathcal{L} = \mathcal{L}_{CE} + \lambda_{KD}\mathcal{L}_{KD}$ as its objective, where $\lambda_{KD}$ is a balancing scalar. In this paper, we investigate logit-based distillation, where $\mathcal{L}_{KD}$ minimises the discrepancy between predicted probabilities by the teacher and the student, and is commonly implemented as the Kullback-Leibler (KL) divergence loss.

### 3.2 Logit-Space Consistency Regularisation

Consistency regularisation has been widely employed in SemiSL research [3, 4, 42, 45, 48]. It involves creating different views of the same unlabelled image, which are separately fed into a neural network to obtain a pair of network predictions. Consistency regularisation is enforced between the pair of predictions given the prior knowledge that both views fundamentally represent the same high-level information such as the object category.

In CRLD, we employ one weak view and one strong to set the stage for our set of within-view and cross-view consistency criterion. Specifically, we adopt RandAugment [13] with random magnitude alongside random crop, random horizontal flip, and Cutout as our strong data augmentation policy. A full list of RandAugment's transformation operations is provided in Supplementary Material. For the weak augmentation, we simply apply random crop and random horizontal flip, which is the standard data augmentation in previous logit-based KD methods [10, 24, 60, 69]. We denote our weak and strong view transformation functions by $T_w(\cdot)$ and $T_s(\cdot)$, respectively.

Concretely, given a batch of $B$ training samples $\mathbf{x} = \{x_b : b \in (1, ..., B)\}$, we separately apply $T_w(\cdot)$ and $T_s(\cdot)$ to each sample to obtain a weakly augmented and a strongly augmented view of $x_b$,

Table 1: Top-1 accuracy (%) on CIFAR-100 with homogeneous-architecture teacher-student pairs.

| Method | | ResNet56 72.34 ResNet20 69.06 | ResNet110 74.31 ResNet32 71.14 | ResNet32×4 79.42 ResNet8×4 72.50 | WRN-40-2 75.61 WRN-16-2 73.26 | WRN-40-2 75.61 WRN-40-1 71.98 | VGG13 74.64 VGG8 70.36 | Avg. |
|---|---|---|---|---|---|---|---|---|
| | Teacher / Student | | | | | | | |
| Feature KD | RKD [38] | 69.61 | 71.82 | 71.90 | 73.35 | 72.22 | 71.48 | 71.73 |
| | FitNets [41] | 69.21 | 71.06 | 73.50 | 73.58 | 72.24 | 71.02 | 71.77 |
| | AT [64] | 70.55 | 72.31 | 73.44 | 74.08 | 72.77 | 71.43 | 72.43 |
| | OFD [23] | 70.98 | 73.23 | 74.95 | 75.24 | 74.33 | 73.95 | 73.78 |
| | CRD [49] | 71.16 | 73.48 | 75.51 | 75.48 | 74.14 | 73.94 | 73.95 |
| | SRRL [56] | 71.13 | 73.48 | 75.33 | 75.59 | 74.18 | 73.44 | 73.86 |
| | ICKD [34] | 71.76 | 73.89 | 75.25 | 75.64 | 74.33 | 73.42 | 74.05 |
| | PEFD [9] | 70.07 | 73.26 | 76.08 | 76.02 | 74.92 | 74.35 | 74.12 |
| | CAT-KD [20] | 71.05 | 73.62 | 76.91 | 75.60 | 74.82 | 74.65 | 74.44 |
| | TaT [32] | 71.59 | 74.05 | 75.89 | 76.06 | 74.97 | 74.39 | 74.49 |
| | ReviewKD [8] | 71.89 | 73.89 | 75.63 | 76.12 | 75.09 | 74.84 | 74.58 |
| | SimKD [6] | 71.05 | 73.92 | 78.08 | 75.53 | 74.53 | 74.89 | 74.67 |
| Logit KD | KD [24] | 70.66 | 73.08 | 73.33 | 74.92 | 73.54 | 72.98 | 73.09 |
| | TAKD [36] | 70.83 | 73.37 | 73.81 | 75.12 | 73.78 | 73.23 | 73.36 |
| | CTKD [31] | 71.19 | 73.52 | 73.79 | 75.45 | 73.93 | 73.52 | 73.57 |
| | NKD [60] | 70.40 | 72.77 | 76.35 | 75.24 | 74.07 | 74.86 | 73.95 |
| | NormKD [10] | 71.40 | 73.91 | 76.57 | 76.40 | 74.84 | 74.45 | 74.60 |
| | DKD [69] | 71.97 | 74.11 | 76.32 | 76.24 | 74.81 | 74.68 | 74.69 |
| | **CRLD** | **72.06** | **74.59** | **78.31** | **76.47** | **75.71** | **75.20** | **75.39** |
| | MLLD † [28] | 72.19 | 74.11 | 77.08 | 76.63 | 75.35 | 75.18 | 75.09 |
| | **CRLD †** | **72.57** | **75.08** | **78.53** | **76.95** | **76.42** | **75.63** | **75.86** |

denoted as $x_b^w$ and $x_b^s$, respectively. Next, we feed both views of the input image individually to the teacher and the student, obtaining four network predictions, namely $\mathbf{p}_w^T$, $\mathbf{p}_s^T$, $\mathbf{p}_w^S$, and $\mathbf{p}_s^S$, where we drop subscript $b$ for brevity.

We define within-view consistency regularisation as the consistency criterion between teacher's and student's predictions of the same weak or strong view. The within-view consistency objective is therefore computed as:

$$\mathcal{L}_{KD}^{WV} = \text{KLD}(\mathbf{p}_w^S, \mathbf{p}_w^T) + \text{KLD}(\mathbf{p}_s^S, \mathbf{p}_s^T) \quad (1)$$

Next, we design a novel cross-view consistency regularisation. It demands the teacher and student to receive differently augmented views of an image and yet produce logit predictions as similar as possible. Formally, this cross-view objective is given by:

$$\mathcal{L}_{KD}^{CV} = \text{KLD}(\mathbf{p}_w^S, \mathbf{p}_s^T) + \text{KLD}(\mathbf{p}_s^S, \mathbf{p}_w^T) \quad (2)$$

The overall KD objective is a sum of the within-view and cross-view consistency losses, that is, $\mathcal{L}_{KD} = \mathcal{L}_{KD}^{WV} + \mathcal{L}_{KD}^{CV}$. Note that although one may tune the weighting of each of the four constituting objective terms, for simplicity we choose to let them share the same weighting in this work. A schematic diagram of the pipeline is provided in Figure 2 .

Furthermore, a previous work [5] reported that teacher and student shall receive an identical view of the same input using the same image transformation for maximal knowledge distillation performance. With our specific design, however, we will demonstrate that *the teacher and student receiving different views of an input using different view transformations leads to optimal performance*. In Section 4.4, we conduct extensive ablation experiments to verify

whether and when the proposed cross-view learning really works. As we will show, cross-view consistency regularisation using different views of the same input image is key to the strong performance of the proposed CRLD framework.

## 3.3 Confidence-based Soft Label Mining

Confirmation bias harms distillation when the student learns from erroneous soft labels provided by the teacher. By introducing the more challenging strongly augmented views, we are also increasing the likelihood that the well-trained teacher produces misleading predictions that undermine student learning. We experimentally observe that strongly-augmented samples generated by our strong view transformation policy can sometimes be almost unintelligible (refer to Appendix for examples), with false predictions made by the teacher. Therefore, we are motivated to refrain unreliable teacher predictions from forming the consistency regularisation pairs. To this end, we propose a simple thresholding mechanism by considering the highest class probability in teacher's per-instance prediction as an indicator of teacher's uncertainty about this prediction. Teacher predictions whose highest class probability is below a given threshold are discarded.

In practice, we apply two thresholds $\tau_w$ and $\tau_s$ for teacher's predictions of the weak and strong views, respectively. Different from the common practice in SemiSL [45, 53], we do not convert the preserved predictions into hard, one-hot pseudo-labels. This is due to the nature of the KD task, whose success hinges on the dark knowledge carried within the non-target class predictions [24, 60, 69]. Instead, we keep teacher's soft predictions as they are as supervision. As an example, objective term $\text{KLD}(\mathbf{p}_w^S, \mathbf{p}_s^T)$ with

**Table 2: Top-1 accuracy (%) on CIFAR-100 with heterogeneous-architecture teacher-student pairs.**

| Method | | ResNet32×4 79.42 ShuffleNetV2 71.82 | ResNet32×4 79.42 WRN-16-2 73.2 | WRN-40-2 75.61 ResNet8×4 72.50 | WRN-40-2 75.61 MobileNetV2 64.60 | VGG13 74.64 MobileNetV2 64.60 | ResNet50 79.34 MobileNetV2 64.60 | Avg. |
|---|---|---|---|---|---|---|---|---|
| | Teacher | | | | | | | |
| | Student | | | | | | | |
| Feature KD | AT [64] | 72.73 | 73.91 | 74.11 | 60.78 | 59.40 | 58.58 | 66.59 |
| | RKD [38] | 73.21 | 74.86 | 75.26 | 69.27 | 64.52 | 64.43 | 70.26 |
| | FitNets [41] | 73.54 | 74.70 | 77.69 | 68.64 | 64.16 | 63.16 | 70.32 |
| | CRD [49] | 75.65 | 75.65 | 75.24 | 70.28 | 69.63 | 69.11 | 72.59 |
| | OFD [23] | 76.82 | 76.17 | 74.36 | 69.92 | 69.48 | 69.04 | 72.63 |
| | ReviewKD [8] | 77.78 | 76.11 | 74.34 | 71.28 | 70.37 | 69.89 | 73.30 |
| | SimKD [6] | 78.39 | 77.17 | 75.29 | 70.10 | 69.44 | 69.97 | 73.39 |
| | CAT-KD [20] | 78.41 | 76.97 | 75.38 | 70.24 | 69.13 | 71.36 | 73.58 |
| Logit KD | KD [24] | 74.45 | 74.90 | 73.97 | 68.36 | 67.37 | 67.35 | 71.07 |
| | CTKD [31] | 75.37 | 74.57 | 74.61 | 68.34 | 68.50 | 68.67 | 71.68 |
| | NormKD [10] | 76.01 | 75.17 | 76.80 | 69.14 | 69.53 | 69.57 | 72.70 |
| | DKD [69] | 77.07 | 75.70 | 75.56 | 69.28 | 69.71 | 70.35 | 72.95 |
| | **CRLD** | **78.58** | **77.38** | **77.70** | **70.98** | **70.59** | **71.56** | **74.47** |
| | MLLD † [28] | 78.44 | 76.52 | 77.33 | 70.78 | 70.57 | 71.04 | 74.11 |
| | **CRLD †** | **78.52** | **77.39** | **77.98** | **71.36** | **70.81** | **71.29** | **74.56** |

**Table 3: Top-1 and Top-5 accuracy (%) on Tiny-ImageNet.**

| Method | | ResNet32×4 64.41/85.07 ResNet8×4 55.25/79.62 |
|---|---|---|
| | Teacher | |
| | Student | |
| Feature KD | FCFD [33] | 60.15/82.80 |
| Logit KD | KD [24] | 56.00/79.64 |
| | DKD [69] | 57.79/81.57 |
| | NKD [60] | 58.63/82.12 |
| | NormKD [10] | 62.05/83.98 |
| | **CRLD** | **63.77/84.57** |
| | MLLD † [28] | 61.91/83.77 |
| | **CRLD †** | **63.84/85.52** |

**Table 4: Top-1 and Top-5 accuracy (%) on ImageNet.**

| Method | | ResNet34 73.31/91.42 ResNet18 69.75/89.07 | ResNet50 76.16/92.86 MobileNetV1 68.87/88.76 |
|---|---|---|---|
| | Teacher | | |
| | Student | | |
| Feature KD | AT [64] | 70.69/90.01 | 69.56/89.33 |
| | OFD [23] | 70.81/89.98 | 71.25/90.34 |
| | CRD [49] | 71.17/90.13 | 71.37/90.41 |
| | CAT-KD [20] | 71.26/90.45 | 72.24/91.13 |
| | SimKD [6] | 71.59/90.48 | 72.25/90.86 |
| | ReviewKD [8] | 71.61/90.51 | 72.56/91.00 |
| Logit KD | KD [24] | 70.66/89.88 | 68.58/88.98 |
| | TAKD [36] | 70.78/90.16 | 70.82/90.01 |
| | DKD [69] | 71.70/90.41 | 72.05/91.05 |
| | NormKD [10] | 71.56/90.47 | 72.12/90.86 |
| | NKD [60] | 71.96/- | 72.58/- |
| | MLLD [28] | 71.90/90.55 | 73.01/91.42 |
| | **CRLD** | **72.05/90.74** | **73.15/91.54** |

confidence-based soft label mining for a batch of training data is mathematically describe as:

$$\mathcal{L}_{KD} = \frac{1}{B} \sum \mathbb{1}(\max \mathbf{p}_s^T > \tau_s) \text{KLD}(\mathbf{p}_w^S, \mathbf{p}_s^T) \qquad (3)$$

where $\mathbb{1}(\cdot)$ is the indicator function. Other objectives are defined like-wise and are omitted for brevity.

## 3.4 Training Objective

The overall training objective for CRLD is a weighted combination of previously described loss terms, namely a ground-truth supervision loss $\mathcal{L}_{CE}$ and $\mathcal{L}_{KD}$.

$$\mathcal{L} = \mathcal{L}_{CE} + \lambda_{KD}\mathcal{L}_{KD} = \mathcal{L}_{CE} + \lambda_{KD}\mathcal{L}_{KD}^{WV} + \lambda_{KD}\mathcal{L}_{KD}^{CV} \qquad (4)$$

where $\lambda_{KD}$ is a balancing weight. $\mathcal{L}_{CE}$ is computed between student's predictions of the weakly-augmented image and the ground-truth label using the cross-entropy loss, which is the standard practice in all previous logits-based distillation methods. $\mathcal{L}_{KD}$ is the sum

of within-view and cross-view consistency regularisation losses described in Section 3.2.

## 4 EXPERIMENTS

### 4.1 Datasets

**CIFAR-100**. CIFAR-100 [29] is a classic image classification benchmark with 50,000 training and 10,000 validation RGB images of 100 classes.

**Tiny-ImageNet**. Tiny-ImageNet is a subset of ImageNet [16]. It consists of 100,000 training and 50,000 validation RGB images over 200 classes, with image resolution downsized from ImageNet's original $256 \times 256$ to $64 \times 64$.

**Table 5: Performance gains by CRLD over diverse logit-based baselines on CIFAR-100.**

| | ResNet56 72.34 | ResNet110 74.31 | ResNet32×4 79.42 | WRN-40-2 75.61 | WRN-40-2 75.61 | VGG13 74.64 | Avg. |
|---|---|---|---|---|---|---|---|
| Teacher | | | | | | | |
| Student | ResNet20 69.06 | ResNet32 71.14 | ResNet8×4 72.50 | WRN-16-2 73.26 | WRN-40-1 71.98 | VGG8 70.36 | |
| KD [24] | 70.69 | 73.57 | 73.53 | 75.22 | 73.74 | 73.43 | 73.36 |
| **+CRLD** | **71.66** | **74.60** | **77.13** | **76.59** | **75.21** | **74.91** | **75.02** |
| NKD [60] | 70.40 | 72.77 | 76.21 | 75.24 | 74.07 | 74.40 | 73.85 |
| **+CRLD** | **71.95** | **74.40** | **78.16** | **76.60** | **74.87** | **75.16** | **75.19** |
| MLLD [28] | 71.24 | 73.96 | 74.64 | 75.57 | 73.97 | 73.80 | 73.86 |
| **+CRLD** | **72.07** | **74.64** | **77.00** | **76.75** | **75.46** | **74.87** | **75.13** |
| DKD [69] | **71.49** | **73.95** | 75.96 | 75.67 | 74.47 | 74.67 | 74.37 |
| **+CRLD** | 70.70 | 73.45 | **77.90** | **76.27** | **75.16** | **75.57** | **74.84** |
| NormKD [10] | 71.43 | 73.95 | 76.26 | 76.01 | 74.55 | 74.45 | 74.44 |
| **+CRLD** | **72.06** | **74.59** | **78.31** | **76.47** | **75.71** | **74.84** | **75.39** |
| MLLD † [28] | 72.19 | 74.11 | 77.08 | 76.63 | 75.35 | 75.18 | 75.09 |
| **+CRLD †** | **72.57** | **75.08** | **78.53** | **76.95** | **76.42** | **75.63** | **75.86** |

**Table 6: Ablation experiments on different designs of consistency regularisation using CIFAR-100.**

| Expt. | $p_w^S$ - $p_w^T$ | $p_s^S$ - $p_s^T$ | $p_w^S$ - $p_s^T$ | $p_s^S$ - $p_w^T$ | ResNet32×4 ResNet8×4 |
|---|---|---|---|---|---|
| 0 | ✔ | | | | 76.26 |
| 1 | | ✔ | | | 76.75 |
| 2 | | | ✔ | | 74.10 |
| 3 | | | | ✔ | 75.38 |
| 4 | ✔ | | ✔ | | 76.57 |
| 5 | ✔ | | | ✔ | 77.58 |
| 6 | ✔ | ✔ | | | 77.76 |
| 7 | ✔ | ✔ | ✔ | | 77.87 |
| 8 | ✔ | ✔ | | ✔ | 78.24 |
| 9 | ✔ | ✔ | ✔ | ✔ | **78.31** |

**Table 7: Ablation experiments student's self-supervised regularisation using CIFAR-100.**

| Expt. | $p_w^S$ - $p_w^T$ | $p_s^S$ - $p_s^T$ | $p_w^S$ - $p_s^T$ | $p_s^S$ - $p_w^T$ | $p_w^S$ - $p_s^S$ | ResNet32×4 ResNet8×4 |
|---|---|---|---|---|---|---|
| 0 | ✔ | | | | | 76.18 |
| 1 | ✔ | | | | ✔ | 72.56 |
| 2 | ✔ | ✔ | | | | 77.86 |
| 3 | ✔ | ✔ | | | ✔ | 77.95 |
| 4 | ✔ | ✔ | ✔ | ✔ | | **78.31** |
| 5 | ✔ | ✔ | ✔ | ✔ | ✔ | 78.16 |

the 30th, 60th, and 90th epochs. Other parameters, unless otherwise stated, follow CIFAR-100 and Tiny-ImageNet experiments.

Our method is implemented in the *mdistiller* codebase, using NormKD [10] as the baseline. Our code and models will be made publicly available for reproducibility.

### 4.3 Main Results

**Distillation performance**. We present extensive experimental results on CIFAR-100, Tiny-ImageNet, and ImageNet datasets using a diversity of teacher-student pairs in Tables 1 to 4. Specifically, the proposed CRLD outperforms all existing methods on all evaluated datasets across teacher-student pairs of both homogeneous (Tables 1, 3, and 4) and heterogeneous (Tables 2 and 4) architectures. When using MLLD's [28] training configurations (marked with "†"), our method achieves further performance gains and leads MLLD by a considerable margin.

**Generalisation capabilities** In addition to NormKD [10], we also apply the proposed CRLD to state-of-the-art logit-based knowledge distillation frameworks [24, 28, 60, 69] and report the results in Table 5. Note that for a fair comparison, we report our reproduced results for compared methods, using official implementations and specifications. The experimental results cogently validate the generalisation capability of our method. The proposed CRLD works orthogonally to existing knowledge distillation methods and can be easily incorporated to significantly boost knowledge transfer

**ImageNet**. A well-known large-scale image classification dataset, ImageNet [16] contains 1.28 million training and 50,000 validation RGB images of 100 classes.

### 4.2 Implementation Details

We evaluate our method by conducting knowledge distillation across various teacher-student pairs of common DNN architecture families: ResNet [21], WRN [63], VGG [44], MobileNet [26, 43], and ShuffleNet [67]. In all experiments, we strictly adhere to standardised training configurations of previous knowledge distillation methods [7–10, 20, 23, 24, 31, 38–41, 49, 50, 55, 56, 60, 64]. All reported experimental results are averaged over 3 independent runs.

**CIFAR-100 & Tiny-ImageNet**. We train our method (and also when reproducing others) for a total of 240 epochs, with an initial learning rate of 0.025 for MobileNet [43] and ShuffleNet [67] students and 0.05 for others. The learning rate decays by a factor of 10 after the 150th, 180th, and 210th epochs; the SGD optimiser is used, with a momentum of 0.9, a weight decay of $5 \times 10^{-4}$, and a batch size of 64.

**ImageNet**. We conduct 100-epoch training with a batch size of 512, with an initial learning rate of 0.2 that decays by a factor of 0.1 at

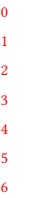

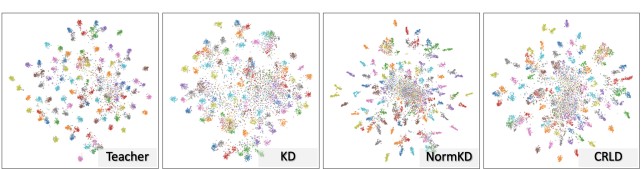

**Figure 3: Evolution of training (top) and test (bottom) set top-1 accuracies (%) on CIFAR-100.**

**Figure 4: t-SNE visualisation of teacher's and distilled student's features on CIFAR-100.**

performance without introducing any extra network parameter or any additional inference overhead.

## 4.4 Ablation Studies

**Design of consistency regularisation.** We break down our full training objective and investigate the play of each individual term in CRLD's overall effectiveness. A set of ablation experiments are conducted with results presented in Table 7. First, we observe that within-view losses are individually effective and consistency within the strong view alone is more effective compared to weak view alone (Expt.0-1). Intriguingly, cross-view consistencies are harmful when used individually (Expt. 1-2), but are rather beneficial when applied in concert with within-view consistencies (Expt. 4-9). Finally, our ablation experiments (Expt. 6-9) demonstrate that each individual consistency objective in our full objective plays a non-negligible part and their joint play leads to the optimal performance. Note that our experiments also highlight that *the effectiveness of CRLD does not stem from a mere increase in the diversity of training samples*, as a notable +1.56% accuracy gain is achieved compared to when the exact same set of strong view augmentation policies are applied in a naive manner (*i.e.*, Expt. 1 →9).

**Self-supervision of student.** A common technique to exploit the potential of unsupervised learning is to enforce a self-supervision criterion on the student. Specifically, the student can receive two different views of the same input and learns to minimise the distance between its predictions of these two views. We investigate whether

**Table 8: Training latency and extra network parameters.**

| Method | Acc. (%) | Latency (ms) | # Param. |
|---|---|---|---|
| FitNets [41] | 73.50 | 21 | 16.8k |
| RKD [38] | 71.90 | 22 | 0 |
| OFD [23] | 74.95 | 26 | 86.9k |
| CRD [49] | 75.51 | 32 | 12.9M |
| ReviewKD [8] | 76.12 | 36 | 1.8M |
| KD [24] | 73.33 | 20 | 0 |
| CTKD [31] | 73.79 | 34 | 52.2k |
| TAKD [36] | 73.81 | - | 270k |
| MLLD [28] | 74.64 | 21 | 0 |
| NormKD [10] | 76.26 | 20 | 0 |
| DKD [69] | 76.32 | 20 | 0 |
| NKD [60] | 76.35 | 20 | 0 |
| **CRLD** | 78.31 | 35 | 0 |

**Table 9: Towards feature-space consistency regularisation.**

| | ResNet32×4 ResNet8×4 | VGG13 VGG8 |
|---|---|---|
| KD (logits) | 73.53 | 73.43 |
| **+CRLD (logits)** | **77.13** | **74.91** |
| FitNets (pool-feat) | 76.74 | 73.87 |
| **+CRLD (pool-feat)** | **77.64** | **75.37** |
| FitNets (feats-3) | 73.66 | **72.27** |
| **+CRLD (feats-3)** | **73.92** | 70.78 |
| FitNets (feats-2) | **73.45** | **72.30** |
| **+CRLD (feats-2)** | 73.17 | 70.37 |

such self-supervision consistency enforce upon the student may bring further improvements to the proposed framework. From the results in Tables 7, we observe that student's self-supervision tends to degrade its own learning. In particular, more severe degradation occurs when less powerful external regularisation *i.e.*, (those imposed by the teacher) are enforced on the student. The same set of ablation experiments on Tiny-ImageNet manifest the same behaviours and are reported in Supplementary Material.Our postulation for such degradation is that the less restricted student, when trained from scratch, is able to find a shortcut in the learnable network parameter space that maximally minimises the distance between its predictions of both views but with poor generalisation. Experimentally, we observe a rapidly converging self-supervision loss quickly dropping to extremely small values, which supports our conjecture. As such, we deem self-supervision regularisation of the student unnecessary and harmful and do not include it in our final optimisation objective.

## 4.5 Further Analyses

**Training curves**. For further insights into the training profile of different methods, in Table 3 we plot the evolution of training and test accuracy at each epoch throughout the training process. We observe that NormKD demonstrates much higher training accuracy than other methods but has only comparable or even lower test accuracy than MLLD and DKD, which implies overfitting on training data. When the proposed CRLD is applied to NormKD, training accuracy lowers while test accuracy notably increases, suggesting

**Table 10: Top-1 accuracy (%) under the label-free knowledge distillation (LFKD) set-up on CIFAR-100.**

| Method | | ResNet56 72.34 ResNet20 69.06 | ResNet110 74.31 ResNet32 71.14 | ResNet32×4 79.42 ResNet8×4 72.50 | WRN-40-2 75.61 WRN-16-2 73.26 | WRN-40-2 75.61 WRN-40-1 71.98 | VGG13 74.64 VGG8 70.36 | Avg. |
|---|---|---|---|---|---|---|---|---|
| | Teacher | | | | | | | |
| | Student | | | | | | | |
| Feature KD | FitNets ‡ [41] | 1.04 | 1.01 | 1.39 | 1.14 | 1.09 | 1.09 | 1.13 |
| | OFD ‡ [23] | 2.09 | 1.13 | 1.43 | 1.49 | 2.27 | 1.71 | 1.69 |
| Logit KD | KD [24] | 70.66 | 73.53 | 73.76 | 74.79 | 73.41 | 73.49 | 73.27 |
| | MLLD [28] | 70.88 | 72.54 | 74.10 | 74.88 | 72.56 | 73.03 | 73.00 |
| | NormKD [10] | **71.36** | **74.35** | 76.49 | 76.04 | 74.82 | 74.39 | 74.58 |
| | **CRLD** | 71.14 | 74.14 | **78.49** | **76.82** | **75.15** | **75.39** | **75.19** |

alleviated overfitting and improved generalisation brought about by CRLD. In addition, we also notice less oscillatory test accuracy curves of CRLD, which is likely due to improved generalisation and mitigated confirmation bias of our method.

**t-SNE visualisation**. We visualise the feature space learnt by the student using different logit-based distillation methods. As seen in Figure 4, features learnt using the proposed CRLD are significantly more seperable in the feature space, with more tightly clustered class-wise features and greater inter-class feature variations. These observations imply greater generalisation of the learnt model and substantiate the superiority of the proposed distillation method.

**Speed & efficiency.** In Table 8, we benchmark the computational efficiency of our method against existing algorithms in terms of average training latency per batch (including data preprocessing) and extra network parameters incurred. Our method has a training latency on par with CTKD [31] and ReviewKD [8] and comparable to some of the fastest methods, and a significant portion of it stems from the on-the-fly data augmentation operations. Besides, our method does not introduce any extra network parameters during training and inference. Note that TAKD [36] involves training additional intermediate assistant networks from scratch and therefore has training costs orders of magnitude higher than others.

**Consistency regularisation in the feature space**. Thus far, we have strictly followed the definition of logit-based distillation by enforcing all consistency regularisation at the network's logit prediction stage. By extension, we are curious about to what extent can our proposed regularisation schemes be extended into the feature space. In theory, closer to the network's input end, features' level of abstraction lowers, and the discrepancy between their representations grows. Forcing feature maps to match can therefore hurt student training within its intermediate stages. Therefore, intuitively we expect degraded student performance as the point of application of consistency regularisation moves towards shallower layers. Shown in Table 9, our experimental results corroborate our intuition. As consistency regularisation moves upper stream — from the logit layer to the pooled feature layer (pool-feat), and then to the 3rd and 2nd feature layers (feats-3 & feats-2), an overall decrease in performance is observed (detailed descriptions found in Supplementary Material). Interestingly, the optimal performance is reached when consistency regularisation is applied to pool-feat instead of the logit space. Nevertheless, the largest performance gains are yielded when it is applied in the logit space, suggesting

that the capacity of consistency regularisation is maximally utilised when working with logits. Motivated by these results, we leave the question of how to unleash the potential of the proposed method within the feature space for future work.

**Distillation without ground-truths.** As consistency regularisation is a widely successful technique in semi-supervised learning, one may naturally ponder whether the proposed CRLD would work with unlabelled training samples. Herein, we formally propose a slightly deviating KD task coined "label-free knowledge distillation (LFKD)", which forbids the use of ground-truth labels during knowledge transfer. LFKD is highly relevant and practical scenario — it is often the case that we have access to only a pre-existing, pre-trained teacher model, but the annotations used to train the teacher are no longer accessible (annotations can be costly not made public especially in industrial contexts). In Table 10, we re-evaluate several methods under the LFKD setting. We find that feature-based methods fail completely under LFKD, which is due to a vague causal linkage between feature mimicking and the downstream classification task. In contrast, logit-based methods deliver comparable performance. CRLD remains as the top-performing method, which signifies its robustness to the absence of annotations. Besides, their performance may be even stronger when using a more knowledgeable teacher with high-accuracy predictions that can largely supersede the role of ground-truth labels. Note that logit-based methods such as DKD and NKD do not support LFKD due to the involvement of ground-truth labels in their objective formulation.

## 5 CONCLUSION

In this paper, we presented a novel logit-based knowledge distillation framework named CRLD. The motivation of CRLD lies in revamping popular ideas found in the semi-supervised learning literature, such as consistency regularisation and pseudo-labelling, to combat the overconfident teacher and confirmation bias problems in knowledge distillation. Our design of within-view and cross-view consistency regularsations, enabled by weak and strong image transformations and coupled with a confidence-based soft label selection scheme, leads to a highly effective and versatile CRLD framework. Extensive experiments demonstrate that CRLD can boost existing logit-based methods by considerable margins and sets new records on different datasets and under different knowledge distillation configurations.

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

Received 20 February 2007; revised 12 March 2009; accepted 5 June 2009

