# OpenReview forum: "Cross-View Consistency Regularisation for Knowledge Distillation"
_acmmm.org/ACMMM/2024/Conference — MM2024 Poster_

### Official Review · Reviewer_kyFE · 2024-05-20

**Rating:** 5
**Confidence:** 3

**Summary:**

This work proposes CRLD, a framework to further advance the capability of knowledge distillation by addressing two long-standing problems in existing KD methods: overconfident teacher and confirmation bias. It introduces extensive within-view and cross-view consistency regularisation to combat the overconfident teacher and over-fitting problems common in KD and a reliable pseudo-label mining module to avoid the negative impact of unreliable and erroneous supervisory signals from the teacher. The proposed framework archives good performance in CIFAR100, TinyImageNet, and ImageNet with numerous teacher-student architectures.

**Strengths:**

1. The writing is clear and sophisticated.
2. The experiments are well-designed, with extensive visualizations, ablations, and settings demonstrated.

**Limitations:**

1. In Section 3.3, this work tries to solve confirmation bias by setting a threshold for confidence-based soft label mining. The proposed metric is termed as $L_{KD}$ in Equation 3, which appears again in Equation 4. However, $L_{KD}$ in Equation 4 is the combination of KLD in Section 3.2, which seems different from it. Would the authors further clarify this?

2. Again in Section 3.3, it would be better if the authors could further clarify how Equation 3 works to be part of the process of knowledge distillation. This part seems a bit confusing.

**Suitability:**

2

---

### Official Review · Reviewer_PjvW · 2024-05-22

**Rating:** 3
**Confidence:** 3

**Summary:**

This paper proposes CRLD, a method for knowledge distillation that employs multiple views of a single sample. Specifically, each input sample is transformed into a weak view and a strong view using different data augmentations, and both views are used for predictions by the teacher and student models. Cross-view and within-view distillation are then performed. Experiments on CIFAR-100 and Tiny-ImageNet demonstrate the effectiveness of the method.

**Strengths:**

1. The motivation is clear: to solve the problem of overconfident teachers and confirmation bias.
2. The paper is well-organized and easy to follow.

**Limitations:**

1. The proposed method has been partially explored in existing works. For example, paper [1] uses different samples from the same class for distillation, which is somewhat similar to using different views of the same sample.
2. Paper [2] points out that identical augmentations for inputs of both the teacher and the student outperform using different augmentations, but the authors' conclusion is contradictory. Additionally, paper [3] notes that performance improvements on small-scale datasets do not necessarily transfer to large-scale ones. Since the experiments in this paper are conducted on CIFAR-100 and Tiny-ImageNet, which are relatively small compared to the ImageNet dataset used in paper [2], results on larger-scale datasets should be provided to further prove the effectiveness of the proposed method. A method that only works on toy datasets cannot support real-world scenarios.

[1]. Regularizing Class-wise Predictions via Self-knowledge Distillation, CVPR 2020.

[2]. Knowledge Distillation: A Good Teacher is Patient and Consistent, CVPR 2022.

[3]. Vanillakd: Revisit the power of vanilla knowledge distillation from small scale to large scale, NeurIPS 2023.

**Suitability:**

2

---

### Official Review · Reviewer_LQNM · 2024-05-25

**Rating:** 4
**Confidence:** 3

**Summary:**

This paper proposed a knowledge distillation method with cross-view consistency regularisation. Based on a weak view and a strong view augmentation, the proposed method aims to solve the overconfident teacher and confirmation bias. Experiments on 3 datasets show that the proposed CRLD method slightly outperforms the previous methods.

**Strengths:**

1. The proposed method is simple and easy to implement.
2. Extensive results have shown the effectiveness of the proposed method.

**Limitations:**

1. Overconfidence is an important issue claimed in this paper. Why does data augmentation (within-view and cross-view) help to reduce this problem? If only the weak or strong augmentation is used, how about the performance? Tables 6 and 6 do not illustrate this point well.
2. In Section 3.3, two thresholds tau_t and tau_s are used for soft label mining. How to determine these two parameters?
3. In Equation 4, how to set the parameter lambda? Is the proposed method sensitive to this parameter?

**Suitability:**

3

---

### Official Review · Reviewer_4vxg · 2024-05-27

**Rating:** 3
**Confidence:** 3

**Summary:**

The paper introduces a new knowledge distillation method called Consistency-Regularisation-based Logit Distillation (CRLD). Existing logit-based Knowledge Distillation (KD) methods have rapidly caught up with feature-based methods in terms of performance, but they still suffer from two main issues during the training process: overconfident teachers and confirmation bias. CRLD addresses these issues by introducing intra-view and inter-view consistency regularization, and by improving the quality of signals transferred from the teacher through confidence-based soft label selection, further mitigating the confirmation bias problem. Despite the seemingly simple design of CRLD, it significantly enhances the learning effect of the student model, achieving new state-of-the-art results on CIFAR-100, Tiny-ImageNet, and ImageNet datasets, and performs well across different teacher and student architectures without introducing additional network parameters.

**Strengths:**

1. The paper proposes the CRLD framework, which innovatively addresses some issues in knowledge distillation through intra-view and inter-view consistency regularization as well as confidence-based soft label selection.
2. CRLD has achieved new state-of-the-art results on multiple datasets, indicating that this method can effectively enhance the performance of student models.
3. The CRLD method does not require the introduction of additional network parameters, which allows it to improve performance while maintaining model size.

**Limitations:**

1.The article employs view consistency to address a range of issues and attempts to enhance the reliability of the teacher model. However, it could attempt to better elucidate the source of the improved reliability, further clarifying the motivations behind this research.
2.The implementation of CRLD may be more complex than traditional knowledge distillation methods, as it involves multi-view data processing and regularization. There is a need for further discussion on the relationship between the performance gains and the increase in computational complexity.
3.The language quality of the article could be improved; for example, the correct spelling in the title should be "Regularization" instead of "Regularisation."
4. The models the author is comparing are rather outdated, it's necessary to include the latest models for comparison.
5. The model baseline used by the author appears to have certain issues, which may be unfavorable for comparison. Further examination is advised.

**Suitability:**

2

---

### Meta-Review · Area_Chair_JVNt · 2024-06-29

**Recommendation:** Accept (Poster)
**Confidence:** 5

**Metareview:**

This work introduces within-view and cross-view regularisations and confidence-based soft label selection to standard logit-based distillation frameworks to address issues, resulting in a significant boost in student learning and setting new state-of-the-art results on multiple datasets. Overall, the paper is technically solid and will have high impact on multimodal perception research. Given the resolution of raised concerns and the unanimous positive reviews, the paper is accepted for publication.